# *Vibrio vulnificus*—A Review with a Special Focus on Sepsis

**DOI:** 10.3390/microorganisms13010128

**Published:** 2025-01-10

**Authors:** Marcello Candelli, Marta Sacco Fernandez, Cristina Triunfo, Andrea Piccioni, Veronica Ojetti, Francesco Franceschi, Giulia Pignataro

**Affiliations:** 1Emergency, Anesthesiological and Reanimation Sciences Department, Fondazione Policlinico Universitario A. Gemelli—IRCCS of Rome, 00168 Rome, Italy; marcello.candelli@policlinicogemelli.it (M.C.); marta.sacco01@icatt.it (M.S.F.); cristina.triunfo01@icatt.it (C.T.); andrea.piccioni@policlinicogemelli.it (A.P.); francesco.franceschi@policlinicogemelli.it (F.F.); 2Department of Internal Medicine, UniCamillus International Medical University of Rome, 00131 Rome, Italy; veronica.ojetti@unicamillus.org

**Keywords:** *Vibrio vulnificus*, sepsis, microbiology, virulence factor, antibiotic

## Abstract

*Vibrio vulnificus* (*V. vulnificus*) is a Gram-negative, halophilic bacillus known for causing severe infections such as gastroenteritis, necrotizing fasciitis, and septic shock, with mortality rates exceeding 50% in high-risk individuals. Transmission occurs primarily through the consumption of contaminated seafood, exposure of open wounds to infected water, or, in rare cases, insect bites. The bacterium thrives in warm, brackish waters with high salinity levels, and its prevalence is rising due to the effects of climate change, including warming ocean temperatures and expanding coastal habitats. High-risk populations include individuals with underlying conditions such as chronic liver disease, diabetes, or immunosuppression, which heighten susceptibility to severe outcomes. The pathogenicity of *V. vulnificus* is mediated by an array of virulence factors, including hemolysins, proteases, and capsular polysaccharides, as well as mechanisms facilitating iron acquisition and immune system evasion. Clinical manifestations range from localized gastrointestinal symptoms to life-threatening systemic infections such as septicemia. Rare but severe complications, including pneumonia and meningitis, have also been reported. Treatment typically involves the use of doxycycline in combination with third-generation cephalosporins, although the emergence of multidrug-resistant strains is an escalating concern. Alternative therapeutic approaches under investigation include natural compounds such as resveratrol and the application of antimicrobial blue light. For necrotizing infections, prompt and aggressive surgical intervention remains essential to improving patient outcomes. As global temperatures continue to rise, understanding the epidemiology of *V. vulnificus* and developing innovative therapeutic strategies are critical to mitigating its growing public health impact.

## 1. Introduction

*Vibrio vulnificus* (*V. vulnificus*) is a Gram-negative, motile, halophilic bacillus capable of causing severe and potentially life-threatening infections, particularly in susceptible individuals. The spectrum of illnesses associated with this pathogen spans from mild gastroenteritis to rapidly progressive conditions such as necrotizing fasciitis and septic shock [1]. Sepsis caused by *V. vulnificus* is a potentially fatal condition with a mortality rate exceeding 50%, particularly in patients with underlying comorbidities such as chronic liver disease, diabetes, hemochromatosis, AIDS, malignancies, and other immunocompromised states [2].

Infections can result from the consumption or handling of contaminated seafood, particularly shellfish such as oysters, or through the exposure of open wounds or broken skin to contaminated saltwater or brackish water [3]. There have been reports of individuals contracting infections without direct contact with contaminated water. *V. vulnificus*, which typically thrives in seawater, has now been found in freshwater environments. Infections can occur through indirect exposure, such as insect bites, without direct contact with infected water [4]. A notable clinical case involved a 53-year-old male who contracted *V. vulnificus* following a bee sting. The patient had no history of exposure to seawater, freshwater, or aquatic organisms or products. Bees, which frequently come into contact with freshwater potentially infected by *V. vulnificus*, can transfer the bacteria onto their bodies via static electricity. When a bee stings a human, the bacteria can enter the body through the sting site. Clinicians treating high-risk patients should heighten their vigilance and consider the atypical routes of *V. vulnificus* infection, particularly in those more susceptible to severe cases. Contamination with *V. vulnificus* can be challenging to detect, as an oyster harboring the harmful bacteria appears, smells, and tastes identical to any other oyster [5]. An analysis of 75 cases reported by the FDA between 2002 and 2007 revealed that symptoms appeared within 0 to 7 days (with an average of 2.0 days), regardless of whether the victim consumed a single oyster or 24 oysters, with symptoms typically developing within 24 h. Interestingly, 33% of those who consumed a single oyster died compared to 25% of those who ate 24 oysters [6]. This indicates that the number of oysters consumed does not influence the time to symptom onset or the outcome of the disease. It suggests that ingesting even a single oyster containing a sufficiently high concentration of *V. vulnificus* strains with a virulent genotype may be enough to initiate the disease in predisposed individuals (Figure 1).

*V. vulnificus* is typically found in coastal or estuarine regions worldwide, where water temperatures range from 9 to 31 °C. However, its preferred habitat is more specific, with optimal conditions occurring at water temperatures above 18 °C and salinities between 15 and 25 parts per thousand (ppt) [7]. Notably, salinities of 30 ppt or higher significantly reduce the prevalence of *V. vulnificus* regardless of temperature [8]. As a result, most infections are linked to tropical or subtropical regions. Global climate change, which causes rising water temperatures, may increase the occurrence of *V. vulnificus* infections and influence the global spread of this pathogen (Figure 2) [9]. Despite the increase in cases, the infection rate remains relatively low, which contrasts with the widespread presence of *V. vulnificus* in marine environments. One possible explanation for this disparity is that *V. vulnificus* typically does not cause severe disease in otherwise healthy individuals [10]. However, individuals with pre-existing conditions such as chronic liver disease, diabetes, hemochromatosis, AIDS, cancer, or weakened immune systems are at a significantly higher risk of infection [2,11]. *V. vulnificus* has been classified into biotypes based on their biochemical characteristics, with three biotypes known to cause severe human diseases [12].

Biotype 1 is the most widespread, typically found in salt or brackish waters. It is the most prevalent strain and is responsible for a variety of illnesses, including primary sepsis.Biotype 2 has a more specialized habitat, primarily in saltwater used for eel (genus *Anguilla*) farming in East Asia and Western Europe. Although it is primarily a serious pathogen of eels, it can, on rare occasions, lead to wound infections in humans.Biotype 3 is a hybrid of biotypes 1 and 2. While it can cause severe soft-tissue infections that may require amputation, its mortality rate appears to be under 8% [1]

The aim of this review is to provide a novel contribution by focusing specifically on *V. vulnificus*-induced sepsis, a severe and often fatal condition with an increasing prevalence due to climate change, even in countries where it has not been previously reported. Given its devastating impact on patient outcomes, a focused analysis of its pathophysiology, clinical presentation, and management strategies is essential to inform clinical practice and drive future research.

## 2. Epidemiology

Vibrio bacteria are responsible for an estimated 80,000 illnesses annually in the United States, with approximately a dozen species known to be pathogenic to humans [13]. Among these, *V. parahaemolyticus* is the leading cause of infections, accounting for about 40% of the reported vibriosis cases, followed by *V. alginolyticus* at approximately 20%. Most individuals infected with Vibrio bacteria experience diarrhea, and many may also experience stomach cramps, nausea, vomiting, fever, and chills. One species, *V. vulnificus*, is associated with severe, life-threatening infections. According to the CDC, there are around 150–200 reported cases of *V. vulnificus* infections each year, with a mortality rate of approximately 20%, often resulting in death within 1 to 2 days after the onset of symptoms [14].

Infections caused by *Vibrio vulnificus* have been documented across a wide range of climate zones globally, with cases reported in numerous countries, including Denmark, Sweden, Germany, Spain, Turkey, the Netherlands, Belgium, Israel, Italy, South Korea, Japan, Taiwan, India, Thailand, Australia, and Brazil [15,16,17,18]. This bacterium is commonly found in seafood, where it poses a significant public health risk. Studies have shown that *V. vulnificus* is frequently detected in seafood samples, with varying prevalence rates depending on the region. For instance, 3.5–8% of the seafood samples from Europe have tested positive for *V. vulnificus*, while 2.4% of the shrimp from Southeast Asia are similarly contaminated. In India, as much as 75% of the freshly harvested oysters have been found to harbor the bacterium. Furthermore, in the Gulf of Mexico, a region known for its warm waters, 100% of the oysters collected during the warmer months, from May to October, have tested positive for *V. vulnificus*. These findings underscore the widespread presence of the pathogen in marine environments, particularly in warmer coastal regions, highlighting the global nature of the risk associated with consuming contaminated seafood [19].

“Healthy patients”, defined as those without identifiable risk factors for *V. vulnificus* infection, are believed to account for less than 5% of all the reported cases of primary sepsis in the United States. In contrast, individuals with chronic liver disease or various immunodeficiencies have been found to be up to 80 times more likely to develop primary sepsis compared to healthy individuals. These estimates of the at-risk population are based on the assumption that all individuals with susceptibility factors are truly at risk for infection [3].

The primary risk factor for *V. vulnificus* infection is chronic liver disease, particularly cirrhosis caused by alcoholism or chronic hepatitis infections such as hepatitis B or C [20]. Research indicates that even moderate alcohol consumption, as little as 30 mL per day, can significantly increase the risk of infection [21]. Other risk factors include immunodeficiencies stemming from dysfunction in macrophages and neutrophils, which can occur due to underlying cancers or immunosuppressive treatments (e.g., for cancer or arthritis), as well as conditions like acquired immunodeficiency syndrome (AIDS), end-stage renal disease (particularly in patients receiving parenteral iron), gastrointestinal issues (such as surgery, ulcers, or achlorhydria), diabetes mellitus, and hematological disorders linked to elevated iron levels, such as hemochromatosis. These conditions compromise the immune system and make individuals more susceptible to severe infections caused by *V. vulnificus* [22] (Table 1).

## 3. Pathophysiology

The pathophysiology of *V. vulnificus* infection results from a complex interaction between the bacterium’s virulence factors and the host’s immune response. For the bacterium to cause an infection, it must first survive in a hostile environment.

### 3.1. Mechanism of Survival in the Hostile Human Body

One of the primary challenges for microorganisms is surviving the highly acidic environment of the stomach. *V. vulnificus* overcomes this obstacle through the production of the enzyme lysine decarboxylase, which breaks down lysine to produce cadaverine. Cadaverine plays a crucial role in neutralizing stomach acid and scavenging superoxide radicals. The interplay between these defense mechanisms is essential for the bacterium’s survival in the digestive tract, allowing it to persist and potentially lead to infection [6]. Once *V. vulnificus* reaches the small intestine, it must adapt to the anaerobic conditions in order to thrive. This adaptation is facilitated by the Fumarate and Nitrate Reduction regulatory protein (FNR), a global transcriptional regulator. FNR plays a pivotal role in enabling the bacterium to proliferate in the intestine, allowing it to reach a critical population density necessary for invasion into the systemic circulation. During anaerobic growth, *V. vulnificus* efficiently utilizes glucose, a key carbon source, through FNR-regulated gene expression. Thus, the initial proliferation of *V. vulnificus* in the intestine is a crucial event, setting the stage for the development of lethal sepsis following oral infection [6].

An important virulence factor that *V. vulnificus* utilizes to evade the innate immune response in the digestive tract is its ability to resist the action of defensins and cathelicidins. These compounds are part of the cationic antimicrobial peptide (AMP) family, which plays a key role in the immune defense. In the small intestine, Paneth cells produce two alpha-defensins (HD-5 and HD-6), while epithelial cells produce a beta-defensin, HBD-1, and a cathelicidin, LL-37. Typically, AMPs function by disrupting bacterial cell membranes and inactivating toxins. However, in the case of *V. vulnificus*, HD-5, HD-6, and HBD-1 exhibited no vibriocidal activity nor any ability to inactivate VVH, the main exotoxin produced by the bacterium, in vitro. Only LL-37 demonstrated significant bactericidal activity. Since LL-37 is primarily produced in the large intestine, this could explain why *V. vulnificus* is able to replicate in the small intestine but is limited in its ability to thrive in the large intestine [23]. Another crucial factor for the growth of *Vibrio vulnificus* is its ability to acquire iron, which is essential for its metabolic processes. The bacterium employs two primary mechanisms for iron uptake: the first involves the use of siderophores, molecules that scavenge iron from transferrin, a protein that typically sequesters iron in the host. The second mechanism involves a heme receptor, HupA, which facilitates the uptake of unbound iron by binding to heme, a form of iron found in host tissues. These strategies enable *V. vulnificus* to effectively acquire the iron necessary for its growth and survival within the host, particularly in iron-limited environments such as the human body [24]. Iron serum levels are directly linked to the infectious dose of *V. vulnificus*, as iron promotes the growth of this microorganism, particularly when transferrin saturation reaches 70%. Elevated iron levels in the blood not only enhance bacterial growth but also play a role in the bacterium’s ability to evade the host’s immune response. The primary iron uptake regulator (FUR) in *V. vulnificus* has been shown to inhibit the transcription of smcR, a key quorum-sensing (QS) regulatory factor. This disruption of quorum sensing affects the entire iron uptake process, enabling *V. vulnificus* to more effectively evade immune detection and promote its survival and proliferation in the host [19]. The importance of iron uptake for *V. vulnificus*’ survival and growth is further highlighted by studies comparing the survival of wild-type and hepcidin knockout mice. Hepcidin is a peptide secreted by hepatocytes that plays a critical role in regulating iron levels by promoting the degradation of ferroportin, the sole known cellular iron exporter. By binding to ferroportin, hepcidin decreases blood iron levels, effectively making this essential element less available to pathogens. This mechanism is crucial during infections, as IL-6, a key pro-inflammatory cytokine, upregulates hepcidin production, causing its levels to increase dramatically during infectious states. The study revealed that while *V. vulnificus* can grow even with limited iron availability, hepcidin knockout mice were significantly more susceptible to infection and exhibited higher mortality rates compared to wild-type mice. Furthermore, treatment with mini-hepcidin PR 73, a synthetic hepcidin agonist, provided protective effects, reducing mortality and suggesting that enhancing hepcidin levels may offer therapeutic potential in mitigating the severity of *V. vulnificus* infections. [25,26]. The observed significant correlation between serum ferritin levels and the survival of *Vibrio vulnificus* is also of particular interest. While there is no direct evidence suggesting that ferritin itself enhances bacterial growth, this possibility cannot be entirely excluded. On the other hand, research by Lien-I Hor et al. found strong associations between elevated serum ferritin concentrations and increased levels of serum AST (aspartate aminotransferase) and ALT (alanine aminotransferase), both of which serve as markers of hepatocellular necro-inflammation. This suggests that elevated ferritin levels might reflect underlying liver damage. Consequently, it is plausible that some other unidentified factors released from damaged liver cells could contribute to enhanced *V. vulnificus* growth in the bloodstream. This finding highlights the complex interaction between the host’s immune response, liver function, and bacterial proliferation during infection [27].

### 3.2. Virulence Factors

In addition to the mechanisms that enhance its survival within the human body, *V. vulnificus* has developed several virulence factors that allow it to damage host cells and evade the immune response. The most important of these are two exotoxins (VVH and MARTX), a protease (VVP), and the polysaccharide component of its capsule (CPS).

#### 3.2.1. *V. vulnificus* Hemolysine (VVH)

*V. vulnificus* hemolysin (VVH) is a water-soluble, thermally unstable extracellular hemolysin and belongs to the cholesterol-dependent cytolysin (CDC) family of pore-forming toxins. It is considered cholesterol-dependent because cholesterol can inactivate VVH by inducing the oligomerization of the toxin monomer. While cellular cholesterol is not a direct receptor for VVH, it is thought to act as a trigger for conformational changes in the toxin, facilitating its transition from a membrane-bound form to a pore-forming state. This ability to form pores in host cell membranes is central to the bacterium’s capacity to cause cell damage and contribute to the progression of infection [28].

VVH exerts its cytotoxicity by binding to and aggregating on the host cell membrane, leading to the formation of small pores. These pores disrupt cellular integrity, causing a variety of cellular outcomes, including necrosis, apoptosis, pyroptosis, and cell lysis across multiple host cell types. The primary targets of VVH include red blood cells, from which the bacterium acquires iron through the release of hemoglobin, as well as endothelial cells, mast cells, and macrophages. The damage caused to these cells is not only central to the bacterium’s ability to invade tissues but also plays a significant role in evading the host’s immune defenses. This is particularly important since macrophages are crucial in the early defense against *V. vulnificus*, even more so than neutrophils. By targeting and damaging macrophages, VVH undermines one of the key components of the host’s immune response, aiding in the bacterium’s ability to persist and spread within the host [29].

In macrophages, even low doses of VVH are enough to induce a significant increase in free cytosolic calcium (Ca^2^⁺), which activates key inflammation-related signaling pathways, including NF-κB, MAPKs, and AKT. The activation of these pathways produces pro-inflammatory cytokines, which are central to the inflammatory response during infection. The release of cytokines plays a fundamental role in the pathogenesis of sepsis, contributing to the systemic inflammatory response and the associated tissue damage and organ failure seen in severe *V. vulnificus* infections. This ability to modulate the host immune response further enhances the bacterium’s virulence and capacity to cause severe, life-threatening illness [18]. Some studies have shown that VVH can also induce necroptosis in mouse macrophages through the Rip1/MLKL pathway. Necroptosis is a form of programmed cell death that is independent of caspase activation, distinguishing it from apoptosis. Unlike apoptosis, necroptosis results in cell rupture and the release of cellular contents, which can trigger a robust inflammatory response. This inflammatory response is particularly concerning in the context of sepsis, as it contributes to tissue damage and organ dysfunction. The activation of necroptosis by VVH may, therefore, play a critical role in the progression of *Vibrio vulnificus*-induced sepsis, amplifying the inflammatory cascade and exacerbating the severity of the infection [30]. Moreover, VVH appears to play a crucial role in the invasion of the bloodstream by *Vibrio vulnificus* through its disruption of the intestinal epithelial barrier. VVH induces NF-κB-dependent mitochondrial cell death, enhances autophagy activation, and promotes paracellular permeabilization in the intestinal epithelium. These effects compromise the integrity of the intestinal wall, facilitating the pathogen’s passage into the bloodstream and the development of septicemia. The small intestine is recognized as the site of the most severe tissue necrosis in *V. vulnificus*-infected individuals, as confirmed by autopsy findings from affected patients. This localized tissue damage in the intestine is a key step in the bacterium’s ability to spread systemically, contributing to the rapid progression of sepsis [31].

Finally, a study in rats found that VVH can induce the dilation of the thoracic aorta by activating guanylate cyclase, which leads to the increased production of cyclic GMP. This mechanism contributes to the development of hypotension, a hallmark of septicemia, sepsis, and septic shock caused by *Vibrio vulnificus* in vivo. The dilation of blood vessels and subsequent drop in blood pressure exacerbates the circulatory dysfunction seen in severe *V. vulnificus* infections, further complicating the clinical management of patients with septic shock. This finding underscores the broader systemic effects of VVH and its role in the pathophysiology of *V. vulnificus*-induced sepsis [32].

#### 3.2.2. Multifunctional Autoprocessing Repeats-in-Toxin (MARTX)

The multifunctional autoprocessing repeats-in-toxin (MARTX) is another critical exotoxin produced by *Vibrio vulnificus* and a key virulence determinant. Encoded by the *RtxA1* gene, MARTX contains four functional domains that contribute to its pathogenic effects [33]

The Domain of Unknown Function 1 (DUF1): This domain acts as a RID-dependent transforming NADase domain (RDTND), playing a role in the modification of host cell functions by cleaving NAD^+^ and impacting cellular metabolism.The Rho Inactivation Domain (RID): This domain inactivates Rho family proteins, which are essential for regulating the cytoskeleton. By damaging the structure of the cytoskeleton, RID contributes to the collapse of the host cell’s internal scaffolding, which disrupts cellular processes and has been linked to the inhibition of MAPK signaling pathways.The α/β Hydrolase Domain (ABH): This domain is involved in hydrolyzing ester bonds, contributing to the breakdown of host cell components and the modulation of cellular responses during infection.The Makes Caterpillars Floppy-like Effector (MCF): Containing a cysteine protease domain (CPD), MCF interacts with unidentified host targets, resulting in Golgi dispersion and cell shrinking. This action further disrupts the structure and function of the infected host cells, promoting the bacterium’s ability to spread and evade immune responses.

Together, these domains make MARTX a potent and multifaceted virulence factor, facilitating *V. vulnificus*’s ability to manipulate host cell signaling, structure, and function, thereby promoting infection and systemic spread.

These domains of the MARTX toxin serve diverse functions that significantly enhance *V. vulnificus*’s ability to grow and invade the host’s bloodstream. One of the key actions of MARTX is its ability to eventually lyse host cells through pore-forming activity, which provides *V. vulnificus* with essential nutrients for its survival and proliferation. The formation of pore-like structures in the plasma membranes of host cells is largely attributed to the amino- and carboxyl-terminal repeated sequence-containing domains of the toxin. These domains enable the toxin to interact with multiple signaling pathways across different cell types, resulting in varied effects that can be either beneficial or detrimental to the host’s immune response. In gut epithelial cells, MARTX interrupts inflammatory responses by inhibiting NF-κB and MAPK signaling pathways. This interference is largely mediated by the RDTND-RID effector duet, which depletes NAD(P)^+^ levels and suppresses the production of pro-inflammatory cytokines like TNF-α. This inhibition of cytokine production reduces the mobilization of Ca^2+^ and paralyzes the early innate immune defense mechanisms, making it easier for the bacterium to persist and spread within the host. In contrast, MARTX has the opposite effect on immune cells. It upregulates pro-inflammatory pathways, including the activation of receptors and signaling molecules such as TLR8, TLR9, CXCR4, STAT4, IKBKB, and IRF7. This dual action creates a paradox in which *V. vulnificus* can evade early immune responses in epithelial cells while simultaneously triggering an exaggerated immune response in immune cells. This uncontrolled immune activation can contribute to systemic inflammation and sepsis that are characteristic of severe *V. vulnificus* infections, ultimately leading to tissue damage, organ failure, and death [34,35].

#### 3.2.3. *Vibrio vulnificus* Protease (VVP)

Another significant enzyme produced by *V. vulnificus* is the VVP, which plays a crucial role in facilitating the bacterium’s dissemination throughout the host’s body. VVP contributes to the progression of infection by causing localized tissue damage and aiding the pathogen’s ability to invade surrounding tissues. Acting as an elastase, VVP degrades structural proteins like elastin in the extracellular matrix, weakening host tissues and enhancing bacterial movement.

One of VVP’s key effects is promoting the production of bradykinin, a peptide that increases vascular permeability. Elevated vascular permeability can lead to the accumulation of fluid in tissues, causing edema and further compromising the host’s local defense mechanisms. This tissue damage, combined with increased vascular permeability, sets the stage for bacterial proliferation and dissemination. In addition to these effects, VVP has been shown to activate prothrombin, leading to clot formation. The resulting microvascular thrombosis can exacerbate tissue ischemia and necrosis, further facilitating bacterial growth in oxygen-deprived environments. Simultaneously, VVP undermines the intestinal barrier by inhibiting the transcriptional expression of Muc-2, a critical mucin produced by goblet cells in the gut. Muc-2 plays a vital role in maintaining the integrity of the intestinal epithelial wall and providing a protective mucosal layer. By disrupting Muc-2 production, VVP contributes to the breakdown of the intestinal barrier, enabling *V. vulnificus* to penetrate the bloodstream and spread systemically [6,36]. Miyoshi et al. demonstrated that VVP, along with its derivative forms, possesses the ability to act directly on various biologically significant human plasma proteins, even in the presence of alpha-macroglobulin, the sole known plasma inhibitor of native VVP. This finding underscores the potency of VVP as a virulence factor, highlighting its capacity to circumvent the body’s natural defense mechanisms. By targeting and degrading essential plasma proteins, VVP disrupts the delicate balance of the proteinase-proteinase inhibitor systems within the human plasma. This imbalance compromises the host’s immune response, potentially leading to an immunosuppressed state. Such a condition diminishes the host’s ability to mount effective defenses against invading pathogens, creating an environment that facilitates the progression of systemic *V. vulnificus* infections, including septicemia [37].

The ability of VVP to evade inhibition by alpha-macroglobulin further amplifies its pathological impact, enabling sustained proteolytic activity within the host’s circulatory system. This mechanism not only aids in the dissemination of *V. vulnificus* but also exacerbates the damage to host tissues and immune systems, thereby playing a pivotal role in the development of life-threatening systemic infections. These multifaceted actions of VVP underline its importance as a virulence factor that not only aids in bacterial survival and dissemination but also significantly contributes to the pathogenesis of *V. vulnificus* infections [37].

#### 3.2.4. *MukB* Gene

Once *V. vulnificus* breaches the intestinal barrier and enters the bloodstream, additional genetic mechanisms become critical to its survival and proliferation. Among these, the *MukB* gene plays a pivotal role. *MukB* is the central subunit of the structural maintenance of chromosomes (SMC) complex, originally characterized in *E. coli*, and it is similarly vital for *V. vulnificus*. *MukB* exists as a homodimer, with each unit containing two ATP-binding pockets that are essential for its function [38]. The SMC complex, to which *MukB* belongs, is responsible for maintaining chromosome organization and integrity during cell division. In the context of *V. vulnificus* infections, *MukB* supports the bacterium’s rapid proliferation within the host’s systemic circulation [29]. This capability is particularly important in the bloodstream, where *V. vulnificus* faces intense immune surveillance and physiological stresses. By ensuring genomic stability and enabling efficient cell division, *MukB* contributes to the pathogen’s ability to adapt and thrive in the host environment. This rapid proliferation is a critical factor in the development of septicemia, as it allows *V. vulnificus* to achieve the high bacterial loads often observed in severe infections. The role of *MukB* highlights the sophisticated strategies employed by *V. vulnificus* to maintain its survival and pathogenesis during systemic infections.

#### 3.2.5. Capsule Polysaccharide (CPS)

The CPS is one of the most critical virulence factors of *Vibrio vulnificus*, playing a crucial role in the bacterium’s ability to evade the host’s innate immune defenses [39]. The CPS provides *V. vulnificus* with antiphagocytic properties, preventing engulfment and destruction by immune cells such as macrophages and neutrophils. Additionally, CPS enhances resistance to complement-mediated killing, a key mechanism of the innate immune system aimed at eliminating invading pathogens. Interestingly, the CPS not only shields the bacterium from immune attack but also contributes to the inflammatory response associated with *V. vulnificus* infections. Research indicates that the encapsulated strain of *V. vulnificus* triggers significantly higher production of pro-inflammatory cytokines, particularly interleukin-8 (IL-8), compared to non-encapsulated strains. This effect is mediated through the enhanced activation of NF-κB transcriptional activity, a central regulator of the inflammatory response. The excessive cytokine production provoked by the CPS contributes to the pathogenesis of septic shock, a severe complication of *V. vulnificus* infections characterized by systemic inflammation, organ dysfunction, and potentially fatal outcomes. By simultaneously evading immune defenses and exacerbating inflammatory responses, the CPS demonstrates a dual role in supporting bacterial survival and driving the progression of disease [39]. This underscores its importance as a target for therapeutic interventions aimed at mitigating the severe outcomes of *V. vulnificus* infections.

#### 3.2.6. The Quorum-Sensing (QS) System

The functionality of *Vibrio vulnificus*’ virulence factors relies heavily on their precise regulation by the bacterium. Central to this regulatory framework is the QS system, a sophisticated communication network that enables bacterial populations to coordinate gene expression based on their density. This system is particularly important for controlling the timing and level of virulence factor production, ensuring that the bacterium’s pathogenic arsenal is deployed optimally during infection. A key component of the QS system is its ability to regulate transcriptional regulators, most notably SmcR. SmcR serves as a master regulator, orchestrating the expression of numerous genes involved in *V. vulnificus* virulence. By activating SmcR and other downstream regulators, the QS system enhances the bacterium’s ability to cause disease. For example, the QS-driven activation of SmcR promotes the synthesis of the Capsule Polysaccharide (CPS), a critical factor for immune evasion and inflammation. This regulation ensures that CPS production aligns with the bacterium’s needs during different stages of infection, maximizing its protective and inflammatory roles. Thus, the QS system is not merely a passive signal transduction mechanism but an active regulator of *V. vulnificus* virulence. Its ability to fine-tune the expression of factors like CPS highlights its importance in the pathogen’s lifecycle, presenting potential targets for interventions aimed at disrupting these regulatory pathways to mitigate *V. vulnificus*-related infections [40].

#### 3.2.7. cAMP-cAMP Receptor Protein (CRP) System

Another critical regulatory system influencing the virulence of *V. vulnificus* is the cAMPCRP system. This system functions as a global regulator, with CRP acting as a transcriptional activator or repressor depending on the target gene. CRP is activated by cyclic AMP (cAMP), which binds to the protein, facilitating its interaction with specific DNA sequences to modulate gene expression.

The cAMP-CRP system plays a fundamental role in coordinating both the bacterium’s metabolic processes and its pathogenic mechanisms. For instance, CRP directly regulates the expression of key virulence genes, including those encoding hemolysin, metalloprotease, and components of the iron uptake system. These factors are vital for the bacterium’s ability to cause disease [41]:Hemolysin: The regulation of hemolysin by CRP ensures optimal timing and levels of this pore-forming toxin, which is critical for the bacterium to lyse host cells, obtain iron, and evade immune responses.Metalloprotease (VVP): CRP control over metalloprotease expression helps the bacterium degrade host tissues, enhance nutrient acquisition, and breach barriers like the intestinal epithelium.Iron Uptake System: Iron is a crucial nutrient for *V. vulnificus*, and CRP facilitates the expression of the genes involved in iron acquisition, including those for siderophores and heme receptors. This regulation ensures the bacterium thrives even in iron-limited environments, such as the bloodstream.

In addition to its role in virulence, the cAMP-CRP system integrates the bacterium’s metabolic state with its pathogenic potential, allowing *V. vulnificus* to conserve energy while efficiently exploiting host resources. This dual role confirms the significance of the cAMP-CRP system as a potential therapeutic target. Disrupting CRP activity could impair both the metabolic adaptability and virulence of *V. vulnificus*, offering a strategy to mitigate infections caused by this formidable pathogen.

#### 3.2.8. The Small Regulatory Protein HlyU

The small regulatory protein HlyU is another crucial virulence regulator in *V. vulnificus*, playing a pivotal role in the bacterium’s pathogenicity. HlyU functions as a transcriptional activator and is essential for the expression of key virulence genes, including those encoding *V. vulnificus* hemolysin (VVHA) and the multifunctional autoprocessing repeats-in-toxin (*RtxA1*) [42]:Upregulation of VVHA:
○VVHA encodes *V. vulnificus* hemolysin (VVH), a pore-forming exotoxin that contributes to host cell lysis, tissue damage, and immune evasion.○HlyU activation ensures sufficient expression of this toxin during infection, allowing the bacterium to damage host cells, release nutrients like iron, and weaken the immune defense.Upregulation of *RtxA1*:
○*RtxA1* encodes the MARTX toxin, a multifunctional effector involved in disrupting host cell cytoskeletal structures, modulating immune responses, and promoting bacterial invasion into the bloodstream.○By regulating *RtxA1*, HlyU enhances the bacterium’s ability to evade immune defenses and establish systemic infections.

#### 3.2.9. AphB

The final virulence regulator of *V. vulnificus* is AphB, a member of the LysR-type transcriptional regulator (LTTR) family, which plays a multifaceted role in enhancing the bacterium’s pathogenicity. AphB is critical for the pathogen’s survival and ability to cause infection, as it influences motility, adherence, immune modulation, and metabolic regulation [34]. AphB regulates the expression of genes that control flagellar synthesis and function, directly impacting the bacterium’s motility. Enhanced motility is crucial for the colonization of the intestinal mucosa. Additionally, AphB modulates the expression of adhesion factors, facilitating the attachment of *V. vulnificus* to host epithelial cells, which is a critical initial step for successful infection. AphB has been shown to activate NF-κB-dependent pathways in intestinal epithelial cells. This activation can lead to the production of pro-inflammatory cytokines such as IL-8, which may contribute to localized inflammation. However, excessive immune activation can disrupt epithelial integrity, aiding bacterial invasion into deeper tissues and the bloodstream. AphB regulates the genes involved in the utilization of carbon sources and other essential nutrients, optimizing bacterial growth and survival under varying environmental conditions. This metabolic flexibility allows *V. vulnificus* to adapt to the nutrient-deprived environments encountered during infection, particularly in the host bloodstream or tissues [43].

### 3.3. Host–Pathogen Interaction

The pathogenesis of *V. vulnificus*-associated illnesses, particularly sepsis and septic shock, is a complex interplay between the pathogen’s virulence factors and the host’s immune response. While much attention has been given to the bacterium’s ability to cause severe tissue damage and evade immune defenses, the host’s immune system also plays a pivotal role in determining the progression and severity of the disease.

#### 3.3.1. Endothelium

Traditionally, the endothelium is seen as a protective barrier and an active participant in immune defense mechanisms. During sepsis, it recruits leukocytes to the infection site, promotes localized coagulation to prevent the spread of pathogens, and releases inflammatory mediators to amplify the immune response. However, in *V. vulnificus*-induced septicemia, the role of the endothelium appears to be a double-edged sword [44]:Triggering Cytokine Storm (CK Storm):
○Interaction between live *V. vulnificus* bacteria and endothelial cells has been proposed as a key event in the development of a cytokine storm.○A cytokine storm is characterized by the excessive and dysregulated release of pro-inflammatory cytokines such as IL-6, TNF-α, and IL-1β, which can lead to systemic inflammation, tissue damage, and multiorgan failure.Endothelial Dysregulation:
○Instead of merely containing the infection, the endothelium may inadvertently contribute to the systemic spread of the pathogen and exacerbate the septicemic process.○This occurs through mechanisms such as increased vascular permeability, which facilitates the dissemination of bacteria and bacterial toxins into the bloodstream.Amplification of Inflammation:
○The interaction between *V. vulnificus* and endothelial cells may further amplify the immune response by activating pathways like NF-κB, leading to the production of pro-inflammatory mediators.○The excessive activation of these pathways can result in a feedback loop, driving the immune system into a hyperactive and damaging state.

In a recent study, researchers investigated the interaction between two strains of the two primary phylogenetic lineages of *V. vulnificus* and human cell lines to better understand the early events of septicemia. Lineage I, which predominantly includes biotype 1 strains, represents the most clinically significant group, as it contains the majority of strains associated with primary septicemia. In contrast, Lineage II includes mostly environmental strains from all three biotypes, along with a smaller subset of strains linked to wound infections and secondary sepsis cases, particularly in aquaculture contexts. The experimental setup used an in vitro infection model designed to mimic the early stages of septicemia, involving two human cell lines: monocytes (MCs)—blood cells crucial for pathogen recognition and immune response—and vascular endothelial cells (VECs), which are key players in inflammatory processes within the vasculature. Real-time PCR analysis revealed that *V. vulnificus* induced the expression of pro-inflammatory cytokines in both MCs and VECs. MCs secreted TNF-α in a dose- and time-dependent manner in response to infection by both *V. vulnificus* strains, with no significant differences observed between the strains. VECs produced and secreted substantial amounts of IL-8 into the culture supernatant following infection. Both strains of *V. vulnificus* caused significant dose- and time-dependent damage to MCs, as evidenced by increased lactate dehydrogenase (LDH) release into the cell culture supernatant. Similarly, the bacteria induced a significant degree of cell damage in VECs when compared to resting, unstimulated cells. These findings highlight the ability of *V. vulnificus* to elicit a robust inflammatory response and induce cellular damage in critical host cell types, underscoring its virulence during the early stages of systemic infection [45]. These results suggest that VECs play a critical role in the early stages of triggering the CK storm in *V. vulnificus*-associated sepsis. Not only do VECs activate the production of pro-inflammatory cytokines upon contact with *V. vulnificus*, but their immune response is also pathogen-specific. Interestingly, VECs do not exhibit the same immune activation when exposed to inactivated or nonpathogenic bacteria, highlighting their ability to specifically recognize and respond to the virulent pathogen.

#### 3.3.2. Macrophage Migration Inhibitory Factor (MIF)

Another important mediator in *V. vulnificus*-induced sepsis is macrophage migration inhibitory factor (MIF). MIF regulates the immune response during infection. In *V. vulnificus*-infected human peripheral blood mononuclear cells (PBMCs), MIF binds to the CD74-CD44 receptor complex, initiating a signaling cascade that leads to the production of key pro-inflammatory cytokines such as TNF-α, IL-6, and IL-8. These cytokines are integral to the inflammatory response and contribute to the pathogenesis of sepsis. Moreover, MIF’s signaling pathways also promote cellular proliferation and inhibit apoptosis, further exacerbating the inflammatory response and tissue damage associated with sepsis [46].

### 3.4. Gut Microbiota–Pathogen Interaction

The gut microbiota plays a crucial role in intestinal homeostasis by interacting with both pathogens and the host’s inflammatory and immune mechanisms. These interactions have also been observed during infections with *V. vulnificus*. For example, it has been noted that mice infected with *V. vulnificus* exhibited modifications in their gut microbiota, including an increase in Firmicutes and a decrease in Bacteroidetes. Among the latter, the species most significantly reduced was *Phocaeicola vulgatus* (*P. vulgatus*). It has been discovered that *V. vulnificus* produces a dipeptide, cyclo-Phe-Pro (cFP), to counteract gut commensals. This peptide is capable of killing *P. vulgatus* by disrupting its cell membrane, thereby facilitating intestinal colonization by the pathogen [47]. Another important interaction is between *V. vulnificus* and *lactobacillus* spp., which are able to resist cyclo-Phe-Pro (cFP) since this peptide cannot disrupt their cell membrane. Additionally, *lactobacillus* spp. produce lactic acid, which inhibits the growth of *V. vulnificus*. However, *V. vulnificus* inhibits the growth of *Lactobacillus* spp. through a quorum-sensing signal molecule called Autoinducer-2 (AI-2) with a mechanism not entirely understood. Mice infected with *V. vulnificus* mutant strains lacking the genes that activate AI-2 show a higher abundance of *Lactobacillus* spp. in their gut microbiota compared to mice infected with wild-type strains of *V. vulnificus* [48]. In Figure 3 are summarized the main virulence factors of *V. vulnificus*.

## 4. Clinical Presentation

*V. vulnificus* infections can lead to serious, potentially life-threatening conditions, particularly in individuals with underlying risk factors. Clinical manifestations of infection can be categorized into typical and atypical presentations. Among the most common symptoms are gastrointestinal issues and necrotizing fasciitis, which can progress rapidly to sepsis. Gastroenteritis often occurs after the consumption of contaminated seafood, such as raw oysters. Symptoms initially include nausea, vomiting, and abdominal discomfort, which can quickly escalate to more severe signs, including fever, chills, and skin manifestations. In many cases, these gastrointestinal symptoms may remain localized and self-limiting, leading to a lack of reporting. While these infections typically do not result in systemic shock or localized cellulitis, they are important to recognize due to the potential for rapid deterioration, particularly in immunocompromised or vulnerable populations [2]. However, in some cases, *V. vulnificus* infections can become fatal, particularly when they progress to primary septicemia or necrotizing fasciitis. Typical skin and muscle lesions associated with severe infections include localized erythema, flaky skin, ecchymosis, blood blisters with exudation, necrosis, cellulitis, and the rapid spread of necrotizing fasciitis. These lesions usually arise following direct contact with contaminated seafood or when open wounds are exposed to tainted water, such as during recreational activities in affected coastal areas. The progression of these conditions is often rapid, with patients experiencing severe tissue damage, systemic symptoms, and potential organ failure, making prompt medical intervention critical [38]. Septicemia is a common manifestation of *V. vulnificus* infection. The small intestine, particularly the ileum, is generally considered the primary entry point for the pathogen, with the cecum also being a likely site of entry. Once the bacteria reach these regions, they can rapidly proliferate and invade the bloodstream, leading to systemic infection and the potential for severe complications [2]. Primary septicemia is characterized by bacteremia without an obvious source of infection, typically presenting with a sudden onset of fever and chills. This is often accompanied by vomiting, diarrhea, abdominal pain, and limb discomfort, usually within a week of consuming contaminated seafood. However, in some cases, symptoms can be delayed for up to 14 days. Within the first 24 h after the onset of illness, secondary skin lesions, including cellulitis, bullae, and bruising, may begin to appear, particularly on the patient’s extremities [2]. Patients with primary septicemia may present with signs of septic shock, including systolic blood pressure < 90 mmHg, altered mental status (such as confusion, lethargy, or disorientation), and thrombocytopenia. The fatality rate for primary septicemia is between 60% and 75%. Notably, the development of hypotension within the first 12 h of admission is a particularly poor prognostic indicator, with these patients being twice as likely to die compared to those who maintain normal blood pressure [49]. Other, rarer clinical presentations of *V. vulnificus* infection have been described, including pneumonia, meningoencephalitis, peritonitis, pyogenic spondylitis, endometritis, septic arthritis, spontaneous bacterial peritonitis, endophthalmitis, and keratitis. A notable case of *V. vulnificus* pneumonia with multiorgan failure was reported in Sri Lanka in 2023. A 46-year-old male patient of Indian origin presented to the emergency department with fever, productive cough with yellow sputum, pleuritic chest pain, and tachypnea over a five-day period. He did not exhibit gastrointestinal symptoms such as vomiting or diarrhea. Blood cultures were positive for *V. vulnificus*. The patient, a non-smoker and teetotaler, worked as a welder at a dockyard, which was likely a key factor in the development of his infection [50]. Another atypical presentation of *V. vulnificus* infection is meningoencephalitis. A case was described involving a young, immunocompromised man who developed severe sepsis after exposure to seawater and the consumption of seafood. The patient later developed meningoencephalitis, with *V. vulnificus* being isolated from both his blood culture and cerebrospinal fluid sample [51]. Another very rare presentation of *V. vulnificus* infection is peritonitis. A case was reported involving a 37-year-old woman undergoing continuous ambulatory peritoneal dialysis (CAPD), who presented to the emergency room with general weakness, fever, diarrhea, and abdominal pain. Despite receiving empirical intraperitoneal antibiotics for suspected CAPD-related peritonitis, her fever persisted. On hospital day 3, she developed hemorrhagic bullae on both lower legs. A review of her recent food history revealed that she had consumed raw seafood prior to admission. The patient was diagnosed with necrotizing fasciitis caused by *V. vulnificus*, which was confirmed after an emergency fasciotomy [52]. In Table 2 are described the clinical presentations of *V. vulnificus* infection.

## 5. Diagnosis of *V. vulnificus* Infections

The diagnosis of *V. vulnificus* infections requires a multifaceted approach that integrates clinical assessment, laboratory investigations, and increasingly, advanced molecular diagnostic techniques. Prompt recognition is vital, especially in individuals with a history of exposure to seawater or raw seafood, as well as those with predisposing risk factors [53,54].

### 5.1. Serum Biomarkers

As expected, laboratory findings in patients with *V. vulnificus* infection often reveal a marked left shift in the white blood cell count, along with rising serum creatinine levels. Elevated creatine phosphokinase (CPK) levels are frequently observed in severe cases, particularly those involving necrotizing fasciitis [55]. Nakafusa et al. demonstrated that elevated CPK is a significant early indicator of *V. vulnificus* infection. This biomarker aids clinicians in identifying high-risk patients, as increased CPK levels correlate with muscle damage and tissue necrosis—hallmarks of severe infection. Moreover, CPK measurement is particularly valuable in the early stages, before definitive microbial identification is available. Monitoring CPK levels over time can also provide critical insights into disease progression and the effectiveness of treatment, facilitating informed clinical decision making. Although serological tests for antibodies against *V. vulnificus* are less commonly used in acute cases of sepsis, Lu et al. note their utility in chronic or convalescent phases of infection [56]. While these tests may support epidemiological investigations, they are not recommended for rapid diagnosis of sepsis due to their limited sensitivity during the early stages of infection.

### 5.2. Imaging Techniques

Imaging plays a critical supplementary role in evaluating the extent of *V. vulnificus* infections, particularly in cases involving necrotizing fasciitis or severe soft tissue damage. Ultrasound and computed tomography (CT) scans are frequently employed to assess the depth and spread of the infection, identify abscess formation, and guide surgical interventions when needed. These imaging techniques are especially valuable in cases with clinical uncertainty regarding the severity or extent of tissue involvement. They provide crucial insights into the infection’s progression, such as the presence of gas in tissues—an indicator of severe infection—and assist in surgical planning [56]. While radiographic studies of affected tissues often reveal nonspecific findings such as soft tissue edema and fluid pockets, CT scans and magnetic resonance imaging (MRI) can offer more detailed evaluations [57]. These modalities are particularly useful for identifying complications like abscess formation, gangrene, or organ involvement, which are common in severe cases of *V. vulnificus* sepsis.

### 5.3. Traditional Microbiological Diagnosis

Blood culture remains the gold standard for diagnosing *V. vulnificus* infection, as it enables the isolation and identification of the bacterium. However, the strict growth requirements of *V. vulnificus* can result in negative blood culture outcomes, particularly if the patient has already received antibiotic treatment or if the pathogen is present in low concentrations [56]. Therefore, while blood cultures are essential, they may not always be the most reliable in severe infections, where rapid diagnosis is crucial to initiate timely and appropriate treatment. In addition to blood cultures, other biological samples can be utilized for microbiological cultures, including material extracted from bullae, ecchymoses, abscesses, sputum, peritoneal fluid, or stool. Stool cultures, in particular, require the use of thiosulfate citrate bile salts sucrose (TCBS) agar for the effective isolation of the pathogen [57].

### 5.4. Nucleic Acid-Based Methods

Given the rapid progression of *V. vulnificus*-associated infections toward severe manifestations, there is an increasing need among clinicians for quick and highly specific diagnostic tools. As a result, numerous molecular techniques have been developed and tested for the detection of this bacterium. Polymerase chain reaction (PCR) has emerged as a valuable diagnostic tool for detecting *V. vulnificus*, particularly in cases where traditional culture methods fail to provide results. PCR enables the direct detection of *V. vulnificus* DNA from clinical samples such as blood, tissue, or wound swabs. Real-time PCR is especially beneficial due to its rapid turnaround time and high sensitivity, facilitating earlier diagnosis and the prompt initiation of treatment. Lee et al. analyzed both serum and skin/soft tissue samples (including bullae fluid, swabs, and surgical debridement specimens) using RT-PCR targeting the toxR gene [58]. This technique demonstrated high sensitivity and specificity, particularly with skin and soft tissue samples, even in cases where antibiotics had been administered prior to sample collection. Moreover, Zhu et al. developed a real-time Recombinase Polymerase Amplification (RPA) assay targeting the vvhA gene of *V. vulnificus* [59]. This assay operates at a temperature of 38 °C, eliminating the need for thermal cycling and reducing equipment requirements. Amplification is completed in about 20 min, making it significantly faster than traditional PCR methods. With its high sensitivity and specificity, this assay is an invaluable tool for the rapid diagnosis and detection of the bacterium, particularly in seafood and aquaculture products. Microarray-based methods, which enable the simultaneous detection of multiple pathogens, have also been adapted for identifying *V. vulnificus*. These methods use a collection of probes immobilized on a solid surface to capture target nucleic acids from a sample. By hybridizing labeled DNA or RNA from the pathogen to the probes, microarrays can quickly detect *V. vulnificus* alongside other marine pathogens [60]. The multiplexing capability of microarrays is particularly advantageous in aquaculture where co-infections are common. Advances in this technology have enhanced detection sensitivity and reduced processing times, making it a practical option for high-throughput applications. Another notable contribution comes from Xiao et al. who developed a method combining Recombinase-Aided Amplification (RAA)—a rapid DNA amplification technique that operates at a constant temperature (37–42 °C)—with the CRISPR/Cas12a system [61]. In this approach, Cas12a, guided by a specific CRISPR RNA (crRNA), recognizes the amplified target DNA and, upon binding, activates non-specific single-stranded DNA cleavage, which generates a detectable fluorescent signal. The entire process takes approximately 40 min and has demonstrated high sensitivity and specificity for detecting *V. vulnificus*.

### 5.5. Advanced Biosensor Technology

Advanced biosensor technologies present promising opportunities for the rapid and sensitive detection of *V. vulnificus*. These biosensors use biological recognition elements—such as antibodies, nucleic acids, or enzymes—coupled with transducers to generate measurable signals. For instance, electrochemical biosensors targeting *V. vulnificus* DNA or RNA sequences offer rapid detection with high specificity and sensitivity, often within minutes. Similarly, fluorescence-based biosensors and surface plasmon resonance (SPR) systems can detect bacterial toxins or specific DNA markers [60]. The portability and point-of-care potential of these devices make them particularly valuable in clinical settings. Moreover, biosensors are particularly effective in detecting bacterial presence in complex environments, such as seafood or water samples, where traditional methods may struggle with specificity due to contamination.

### 5.6. Metagenomic Next-Generation Sequencing (mNGS)

Metagenomic next-generation sequencing (mNGS) has emerged as a powerful tool for diagnosing infections like *V. vulnificus*, particularly in cases where traditional diagnostic methods fail. mNGS allows for the comprehensive analysis of microbial DNA from clinical samples, detecting a wide range of pathogens simultaneously, including those that may be present in mixed infections. In particular, Wang et al. highlighted the value of mNGS in identifying pathogens in skin and soft tissue infections, where *V. vulnificus* is often involved, as it provides a broad and rapid diagnostic capacity; mNGS identified *V. vulnificus* in patients with SSTIs at a much higher rate (91.5%) compared to traditional diagnostic methods (60.5%) and was also able to identify mixed infections in which *V. vulnificus* was present alongside other pathogens, which traditional culture methods often missed [62]. This is important because *V. vulnificus* is frequently found in polymicrobial infections, particularly in cases related to marine environments or seafood exposure. Li et al. demonstrated the utility of metagenomic next-generation sequencing (mNGS) in diagnosing *V. vulnificus* infections in patients with negative culture results [63]. In their study, traditional culture methods identified *V. vulnificus* in only 9 of 14 suspected cases (64.3%), leaving a diagnostic gap. In contrast, mNGS detected *V. vulnificus* in five cases that were negative by culture, showcasing its ability to identify pathogens in challenging clinical scenarios. One significant advantage of mNGS is the rapid turnaround time, with results available in 24–48 h, far quicker than traditional culture methods. This is crucial in cases where severe clinical manifestations demand a fast and accurate diagnosis to ensure timely antibiotic treatment, potentially saving the patient’s life. Despite its promise, molecular and sequencing-based techniques are not yet universally accessible due to their cost and infrastructure requirements. However, ongoing technological advancements are expected to improve the availability of these methods, bridging the gap between research and routine clinical practice.

## 6. Prognostic Factor Associated with *V. vulnificus* Infection

Considering how *V. vulnificus*-associated infections, especially bacteremia, primary septicemia, and septic states, can quickly lead to death in susceptible patients, some studies have been performed aiming to find predictive factors for unfavorable out-comes. In an emergency department setting, many clinical scores are already used to predict the outcome of septic and critical patients, such as qSOFA, APACHE II, and MEDS; all these scores can be effectively used to predict the outcome in patients with *V. vulnificus* infections. However, other parameters have also been studied. For instance, it has been demonstrated that serum TNF-α levels are significantly higher in patients with *V. vulnificus* infections versus a healthy control group, and also significantly higher in infected non-survivors versus survivors; at a cutoff value of 100 pg/mL, the sensitivity and specificity for mortality prediction in one study were, respectively, determined to be 91% and 71.4% [64]. A study successfully adapted the qSOFA and MEDS scoring systems to improve the prediction of outcomes in *Vibrio* bacteremia by incorporating additional clinical parameters such as Blood Urea Nitrogen (BUN) and pH. Specifically, if BUN exceeded 25 mg/dL, the modified score added one point, and if pH was lower than 7.36, another point was added. The modified MEDS score, with a cutoff value of 10, demonstrated a sensitivity of 77.8% and specificity of 85.2%. Meanwhile, the modified qSOFA score, with a cutoff value of one, showed a sensitivity of 88.9% and specificity of 59.3%. These modifications enhance the predictive value of these scoring systems for *V. vulnificus* infections, particularly in terms of identifying patients at a higher risk for mortality [65].

## 7. Antibiotic Treatment

In 2023, the US Centers for Disease Control and Prevention (CDC) issued a Health Alert Network regarding severe *V. vulnificus* infections, reaffirming previous treatment recommendations. The standard antibiotic regimen consists of doxycycline (100 mg orally or intravenously twice a day for 7–14 days) in combination with a third-generation cephalosporin (TGC), such as ceftazidime (1–2 g intravenously or intramuscularly every 8 h). Alternative treatment options include pairing a TGC with a fluoroquinolone (e.g., ciprofloxacin 500 mg orally twice a day) or administering a fluoroquinolone alone. For pediatric patients, the recommended regimens include a TGC combined with doxycycline, ciprofloxacin, or trimethoprim-sulfamethoxazole paired with an aminoglycoside [14]. These regimens aim to address the severe nature of *V. vulnificus* infections, particularly those leading to sepsis. Many studies over the years have sought to identify the most effective antibiotic combinations for treating the various clinical manifestations of *V. vulnificus* infection. For example, Jang et al. proposed that the combination of cefotaxime and ciprofloxacin might be more effective than other commonly used regimens, such as those involving minocycline and cefotaxime. This is primarily due to ciprofloxacin’s ability to reduce the transcription of RtxA1, a virulence factor in *V. vulnificus*, thereby decreasing subsequent cytotoxicity in cases of *V. vulnificus*-associated sepsis [66]. In cases of foodborne septicemia, Trinh et al. demonstrated that the most effective treatments include regimens such as ceftriaxone–doxycycline, ceftriaxone–ciprofloxacin, cefepime–doxycycline, and cefepime–ciprofloxacin [67]. Interestingly, doxycycline monotherapy also proved to be beneficial, despite its ineffectiveness in wound infection models. This efficacy may be attributed to doxycycline’s high concentrations in the small intestine due to its excretion in bile [67]. Additionally, since beta-lactams are commonly used in the empirical treatment of septicemia, the researchers tested cefepime and ceftriaxone, confirming data from the Cholera and Other *Vibrio* Illness Surveillance System (COVIS), which suggests that cephalosporins alone are ineffective against *V. vulnificus* [68]. Similarly, in the context of necrotizing skin and soft tissue infections (NSTIs), a regimen of TGC plus ciprofloxacin was found to be as effective as TGC plus doxycycline in a study conducted by Kim et al. [69]. In cases of necrotizing fasciitis where standard regimens have failed, tigecycline has shown a good response, likely due to its ability to reach high concentrations in soft tissues and its cytokine immunomodulatory effects [69]. Tigecycline is a member of the glycylcycline class of antibiotics, approved for use in complicated soft tissue and intra-abdominal infections. Due to its pharmacokinetic profile, which leads to low serum concentrations, tigecycline monotherapy may not be the most effective treatment, particularly in *V. vulnificus* infections, which are often associated with bacteremia. Consequently, a combined regimen of tigecycline and ciprofloxacin could offer a more effective treatment option, as in vitro studies have demonstrated some degree of synergy between these two drugs [70]. Antibiotic therapy alone may be insufficient for treating necrotizing skin and soft tissue infections (NSTIs), particularly necrotizing fasciitis. Prompt surgical intervention is universally regarded as a critical prognostic factor for survival. Specifically, surgical treatment should be initiated within 12 h of admission. Studies have shown that delaying surgery beyond 12 h significantly worsens outcomes. While there is no substantial difference in mortality risk between patients who undergo surgery between 12 and 24 h or after 24 h of admission, the group treated within the first 12 h consistently exhibits better outcomes [71].

## 8. Antibiotic Resistance

It is impossible to overlook the growing concern of multidrug-resistant (MDR) bacteria, driven largely by the indiscriminate use of antibiotics in both hospital settings and in industries such as agriculture and aquaculture. This phenomenon has become a global health crisis, and *V. vulnificus* is no exception. The emergence of MDR strains of *V. vulnificus* poses a significant challenge to effective treatment, complicating efforts to manage infections and increasing the risk of poor clinical outcomes. This underscores the urgent need for stricter antibiotic stewardship and continued research into alternative therapies. *V. vulnificus* has developed a variety of antibiotic resistance genes (ARGs) and produces several types of orthologous proteins, which can be classified into five main resistance mechanisms: (1) antibiotic efflux, where the bacteria actively pump out antibiotics to reduce their intracellular concentration; (2) antibiotic inactivation, involving enzymes that degrade or modify the antibiotic, rendering it ineffective; (3) antibiotic target alteration, where mutations in the target sites prevent the antibiotic from binding; (4) antibiotic target protection, where protective proteins shield the antibiotic target from the drug’s effects; and (5) antibiotic target replacement, where alternative molecules or pathways take over the function of the antibiotic target, allowing the bacteria to bypass the drug’s action [72]. These resistance mechanisms contribute significantly to the bacterium’s survival in the presence of antibiotics, complicating treatment options and posing a major challenge for public health. As a result, there has been an increasing number of reports concerning resistance in *V. vulnificus* across various antibiotic classes. The most implicated antibiotics include ampicillin, penicillin, and tetracycline. However, *V. vulnificus* has also been shown to resist other antibiotics such as aztreonam, streptomycin, gentamicin, and tobramycin [73]. Of particular concern are emerging resistances to last-resort antimicrobials, including carbapenems and third- and fourth-generation cephalosporins. These resistances are often associated with mobile genetic elements, further complicating treatment options. Alarmingly, such resistant strains have been detected in seafood isolates, particularly in imported products, raising concerns about the spread of resistant *V. vulnificus* through global food supply chains [73].

## 9. Other Therapeutic Regimen

In response to the growing challenge of antibiotic resistance and the slow pace of new antibiotic development, numerous studies have explored alternative therapies for *V. vulnificus* infections. One such promising candidate is resveratrol (RSV), a natural compound known for its potential to activate SIRT1, a protein that interacts with molecules involved in necroptosis [30]. Necroptosis, a form of programmed cell death distinct from apoptosis, plays a role in the severe inflammatory response associated with *V. vulnificus* infections. Research has demonstrated that RSV pretreatment can downregulate key necroptosis-related molecules such as RIP1, RIP3, and MLKL in peritoneal macrophages, thereby potentially preventing the onset of sepsis and septic shock. This approach is especially relevant, as the overwhelming inflammatory response to *V. vulnificus* infections is not primarily due to apoptosis, but rather to other cell death pathways, making RSV a valuable candidate for further investigation as an adjunctive therapy. Moreover, resveratrol (RSV) is a natural antioxidant known for its anti-inflammatory properties, including the inhibition of the COX-1, COX-2, and NF-kB pathways. Its effectiveness against *V. vulnificus* may be attributed not only to its antioxidant activity but also to its ability to impair bacterial virulence. Specifically, resveratrol has been shown to reduce swarming motility, prevent adhesion to host cells, and downregulate the production of the RtxA1 toxin, which is crucial for the bacterium’s cytotoxicity. A study demonstrated that pre-treatment with resveratrol at a dose of 20 mg/kg provided significant protection, with 80% of the mice surviving the infection. In contrast, when administered 2 h post-infection, resveratrol still provided protection, albeit to a lesser extent, with 40% of the mice surviving. These results suggest that resveratrol holds promise as a therapeutic agent, particularly when used as a preventive measure, though its potential benefits in post-infection treatment warrant further investigation [74]. Otilonium bromide, a spasmolytic agent traditionally used to treat symptoms of irritable bowel syndrome (IBS), has shown potential as an inhibitor of *V. vulnificus* pathogenicity. Studies suggest that otilonium bromide can prevent bacterial cell division, reduce swarming motility, and decrease adhesion to host cells, all of which are critical factors in the bacterium’s ability to establish infection. It achieves these effects by inhibiting the expression of FlaB (a key component of the flagellar motor) and CRP (a transcriptional regulator), as well as suppressing the secretion of the RtxA1toxin, which contributes to the cytotoxicity of *V. vulnificus*. Moreover, otilonium bromide has demonstrated a synergistic effect when used in combination with antibiotics like tetracycline or erythromycin, enhancing their bactericidal activity. This makes otilonium bromide a promising adjunct to traditional antibiotic therapies, potentially improving treatment outcomes for *V. vulnificus*-induced infections [75]. Fursultiamine hydrochloride (FTH), a thiamine derivative used primarily to treat thiamine deficiency, has shown promising potential as an antimicrobial agent against *V. vulnificus* in vitro. FTH was found to inhibit the transcription of HlyU-regulated toxin genes, specifically rtxA1 and vvhA, which are responsible for the production of pore-forming toxins [76]. By downregulating these toxin genes, FTH reduces the bacterium’s ability to secrete toxins and kill host cells, potentially mitigating the severity of *V. vulnificus* infections. However, when tested in vivo, the instability of FTH led to unsatisfactory results, limiting its current clinical applicability. These findings suggest that while FTH has potential as a therapeutic option, further research and modifications to enhance the drug’s stability and efficacy are needed before it can be considered for clinical use [76]. Bergamottin, a compound abundant in citrus fruits such as bergamot and pomelo, has been studied for its potential therapeutic effects against multidrug-resistant (MDR) *V. vulnificus* infections [77]. As an inhibitor of cytochrome P4501A1 (CYP1A1), bergamottin interferes with pathways that can negatively regulate the body’s immune response to infections. Research by Ruo-Bai Qiao et al. demonstrated that bergamottin administration during lethal *V. vulnificus* infections in mice significantly prolonged survival and reduced organ damage, particularly to the liver and kidneys [77]. Interestingly, bergamottin’s effect was found to be more beneficial than a combination therapy involving gentamicin, cefotaxime, and levofloxacin, which are typically used in the treatment of *V. vulnificus* infections. This protective effect has been attributed to bergamottin’s ability to inhibit the nuclear factor Kappa B (NF-κB) signaling pathway, a critical mediator of inflammation and immune response during infections. These findings suggest that bergamottin may offer a promising adjunctive therapy for treating MDR *V. vulnificus* infections, although further studies are needed to fully understand its mechanisms and therapeutic potential [77]. Mannitol, a commonly used drug primarily known for its osmotic diuretic properties, has also shown some in vitro activity against *V. vulnificus*. Studies indicate that mannitol can downregulate the expression of the genes encoding essential virulence factors such as PlpA, VvpE, and RtxA. These factors are critical to *V. vulnificus*’s ability to cause tissue damage, evade host immune responses, and proliferate in the host. By reducing the expression of these virulence-related genes, mannitol could potentially limit the bacterium’s ability to cause severe infections like necrotizing fasciitis and septicemia [78]. Lastly, an intriguing potential treatment modality for *V. vulnificus* infections is antimicrobial blue light (aBL), specifically with a wavelength of 450 nm [79]. Studies have demonstrated that aBL can effectively kill a wide range of microorganisms through the excitation of endogenous chromophores, such as porphyrins, which in turn induce the production of reactive oxygen species (ROS). These ROS are toxic to the bacteria, leading to their destruction [79]. In one study, researchers found that aBL exposure significantly increased survival rates in mice with skin infections caused by *V. vulnificus*. The treatment may offer a critical advantage by providing a window of time for antibiotics to take effect, potentially improving patient outcomes. However, the effectiveness of aBL appears to be influenced by skin thickness, with male mice showing less favorable results compared to female mice. This variation could pose challenges when considering its application in humans, where skin thickness may vary. Despite these limitations, the promising results suggest that aBL could become a valuable adjunctive therapy, especially in the treatment of skin and soft tissue infections caused by *V. vulnificus* [79]; in fact, the use of aBL and similar approaches, such as femtosecond laser-based therapy, is an increasingly interesting topic, especially when taking into consideration antibiotic-resistant microorganism. For instance, femtosecond laser-based therapy has been studied and successfully tested against *S. aureus*, *P. aeruginosa*, and vancomycin-resistant *E. faecalis*, respectively, at wavelengths from 370 to 420 nm in the first study and from 420 to 465 nm in the second one [80,81]. Further research is needed to optimize the use of aBL and femtosecond laser-based therapy and assess its potential for broader clinical application.

## 10. Potential Clinical Implications and Future Directions

Sepsis and septic shock represent frustrating conditions for doctors. Despite the enormous progress made in recent decades in all fields of medicine (such as the treatment of HIV infection, the development of biological drugs in the field of autoimmunity, and the advances in oncology), the mortality rate for sepsis has remained unchanged [82]. Furthermore, apart from personalizing antibiotic therapy based on susceptibility tests conducted on the responsible microorganism and the mechanical removal of the septic focus when possible, there are no significant differences in treatment based on the causative agent [83]. The study of the pathophysiological mechanisms of *V. vulnificus* infection has been particularly effective, identifying numerous virulence and pathogenicity factors that could lead to clinical advancements and future research. Research should focus on identifying the molecules capable of precisely targeting these factors with specific inhibitors. Another important step could be to improve the understanding of the interaction between the commensal microbiota and *V. vulnificus*. In this regard, a probiotic, chitosan A, has been shown to counteract the growth and infection caused by *V. vulnificus* both in vitro and in animal models through direct toxicity mechanisms and possibly by modulating the gut microbiota [84]. Finally, research should continue on non-pharmacological treatments, such as aBL, which appear to be extremely promising.

## 11. Methods

This review includes studies published over the last 30 years on *V. vulnificus*, with a comprehensive search conducted across Scopus, PubMed^®^, Web of Science^®^, and Cochrane^®^. We reviewed a variety of study types, including literature reviews, observational studies (case–control, cross-sectional), retrospective and prospective studies, clinical trials, and case reports, as well as in vivo and in vitro studies, all in the English language. Data were extracted based on the period of research, title, abstract, and study type. Ethical approval is not required for this review. The primary search terms used were as follows: *V. vulnificus* AND sepsis, *V. vulnificus* AND septic shock AND/OR infection, AND treatment AND/OR pathophysiology, and *V. vulnificus* AND microbiota (Figure 4).

## 12. Conclusions

*V. vulnificus* is a highly virulent pathogen that poses a significant threat, particularly in susceptible individuals, leading to severe infections such as primary septicemia, septic shock, and necrotizing fasciitis. The pathogenesis of these infections involves complex interactions between bacterial virulence factors and the host’s immune response. While many studies have advanced our understanding of *V. vulnificus*’s mechanisms of infection and its ability to evade immune defenses, the rapid progression of septicemia remains a major clinical challenge. Early diagnosis, prompt antibiotic treatment, and, when necessary, surgical intervention are critical for improving patient outcomes. However, the emergence of multidrug-resistant strains highlights the urgent need for novel therapeutic strategies.

## Figures and Tables

**Figure 1 microorganisms-13-00128-f001:**
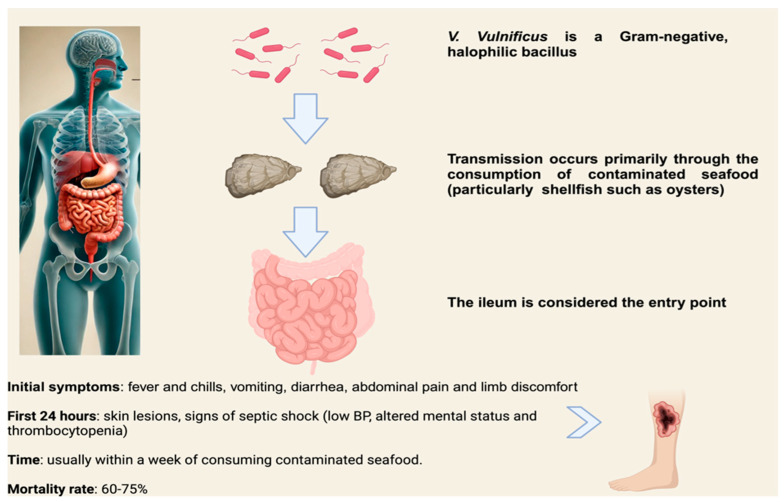
*V. vulnificus* infection. Created in BioRender (https://BioRender.com/f89u105, URL created on 28 December 2024).

**Figure 2 microorganisms-13-00128-f002:**
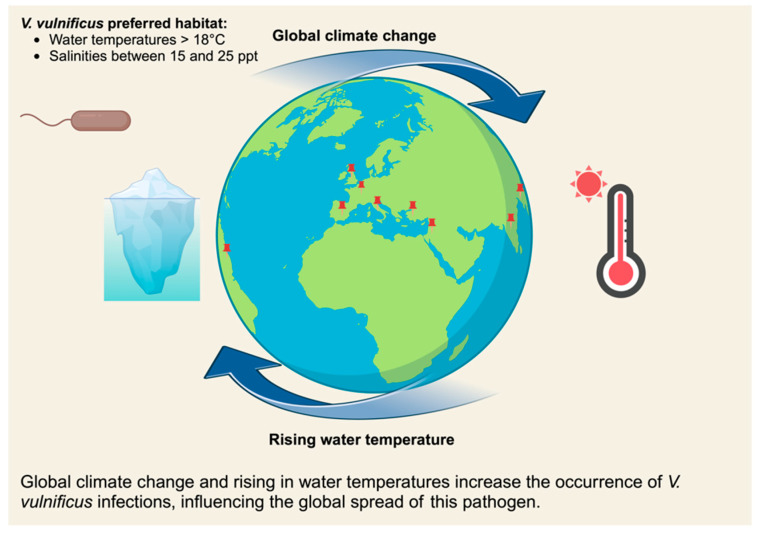
Climate change increased *V. vulnificus* diffusio. Created in BioRender (https://BioRender.com/f89a566, URL created on 28 December 2024).

**Figure 3 microorganisms-13-00128-f003:**
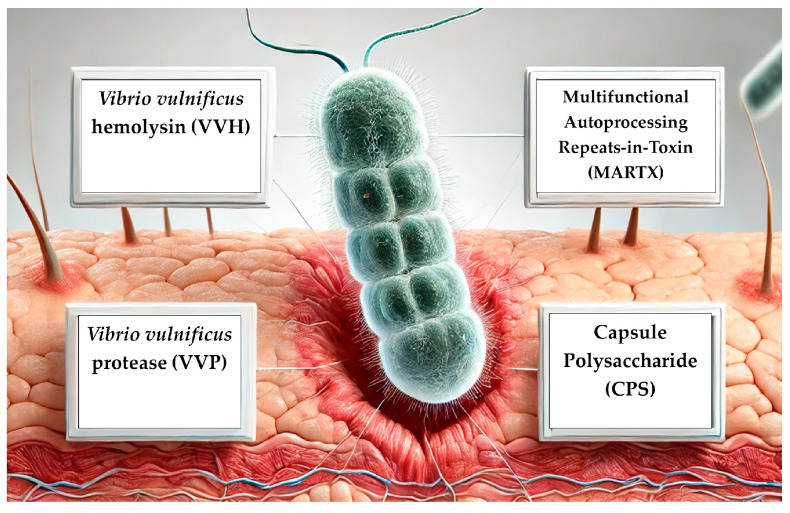
Main virulence factors of *V. vulnificus*.

**Figure 4 microorganisms-13-00128-f004:**
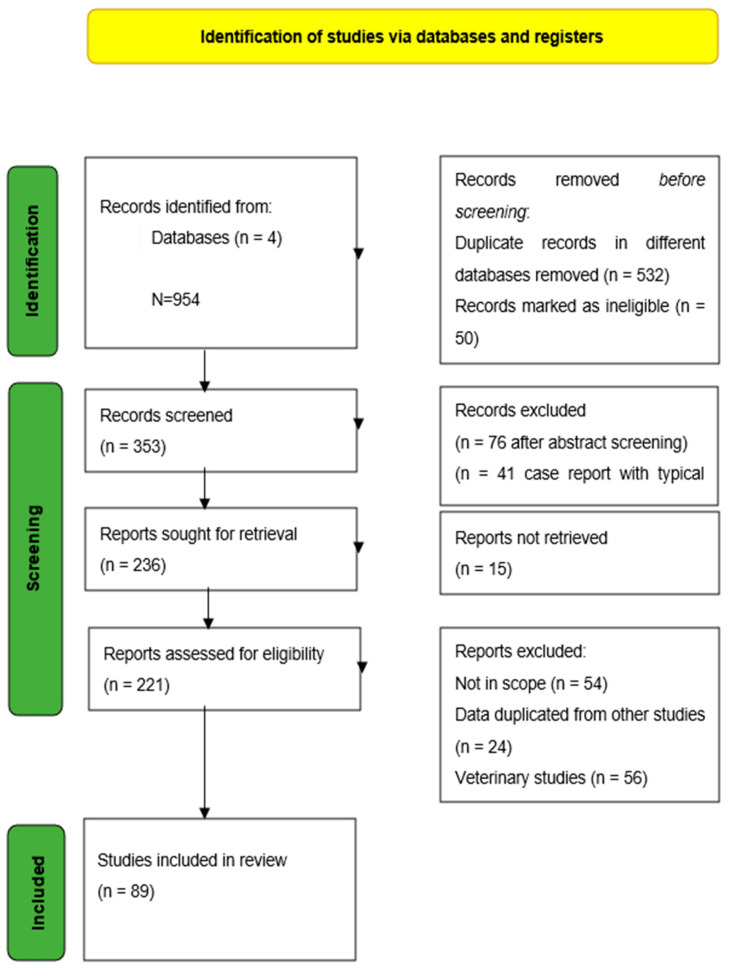
Prisma flow diagram.

**Table 1 microorganisms-13-00128-t001:** Risk factors for *V. vulnificus* infection.

System Involved	Risk Factors
Liver	Chronic liver disease: cirrhosis from alcoholism or chronic hepatitis infections
Kidney	End-stage renal disease
Gastrointestinal tract	Ulcers, previous surgery, and achlorhydria
Pancreas	Diabetes mellitus
Hematological disorders	Hemochromatosis; immunodeficiencies (cancer, immunosuppressive therapies, and AIDS)

AIDS: acquired immunodeficiency syndrome.

**Table 2 microorganisms-13-00128-t002:** Clinical manifestations of *V. vulnificus* infections.

Common Presentations	Rare Presentations
Gastrointestinal symptoms	Pneumonia
Necrotizing fascitis	Meningoencephalitis
Sepsis	Peritonitis
	Pyogenic spondylitis
	Endometritis
	Spontaneous bacterial peritonitis
	Endophtalmitis/keratitis

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
