# Peer review of "Vibrio vulnificus—A Review with a Special Focus on Sepsis"

_microorganisms, 2025, doi:10.3390/microorganisms13010128_

Round 1

Reviewer 1 Report

Comments and Suggestions for Authors

The paper entitled:”VIBRIO VULNIFICUS – A REVIEW with a Special Focus on Sepsis “ by Cancelli M. et al. deals with the Gram-negative, halophilic bacillus involvement in severe infections such as gastroenteritis, necrotizing fasciitis, and septic shock; whose transmission mainly occurs through the consumption of contaminated seafood, exposure of open wounds to infected water, or insect bites. 

Interestingly, the diffusion of this bacterium is linked to climate changes, such as warming ocean temperatures and expanding coastal habitats. 

As the authors point out global temperatures continue to rise, understanding the epidemiology of V. vulnificus and developing innovative therapeutic strategies are critical to mitigating its growing public health impact.

Current treatment typically involves doxycycline in combination with third-generation cephalosporins, although the emergence of multidrug-resistant strains poses an issue here. Some alternative therapeutic approaches have also been covered.

The review includes studies published over the last 30 years (comprehensive search across various biomedical databases, covering reviews, observational studies, retrospective and prospective studies, clinical trials, case reports, in vivo and in vitro studies, and pre-clinical studies) but only 64 bibliographical entries have been cited. This sounds 

As such the review is very comprehensive and useful. However, compared to many other recent papers in the field, such as for instance https://jamanetwork.com/journals/jama/fullarticle/2801603

having more than 170 bibliographic entries and various captivating graphical illustrations.

We understand that this review has a focus on sepsis but we suggest that authors check and improve citations and put at least some more graphical material in the manuscript to help the “at a glance” fruition of the paper.

One about climate changes and increased V.Vulnifucus diffusion could be one.

Another image could deal with V. Vulnificus in sepsis.

How does sepsis general treatment differ from specific pathogen directed treatment?

Author Response

General comment: The paper entitled:”VIBRIO VULNIFICUS – A REVIEW with a Special Focus on Sepsis “ by Candelli M. et al. deals with the Gram-negative, halophilic bacillus involvement in severe infections such as gastroenteritis, necrotizing fasciitis, and septic shock; whose transmission mainly occurs through the consumption of contaminated seafood, exposure of open wounds to infected water, or insect bites. Interestingly, the diffusion of this bacterium is linked to climate changes, such as warming ocean temperatures and expanding coastal habitats.  As the authors point out global temperatures continue to rise, understanding the epidemiology of V. vulnificus and developing innovative therapeutic strategies are critical to mitigating its growing public health impact. Current treatment typically involves doxycycline in combination with third-generation cephalosporins, although the emergence of multidrug-resistant strains poses an issue here. Some alternative therapeutic approaches have also been covered.

Answer: "The authors thank the reviewer for the suggestions provided, which have significantly improved the manuscript"

Comment 1: "The review includes studies published over the last 30 years (comprehensive search across various biomedical databases, covering reviews, observational studies, retrospective and prospective studies, clinical trials, case reports, in vivo and in vitro studies, and pre-clinical studies) but only 64 bibliographical entries have been cited. This sounds As such the review is very comprehensive and useful. However, compared to many other recent papers in the field, such as for instance https://jamanetwork.com/journals/jama/fullarticle/2801603 having more than 170 bibliographic entries and various captivating graphical illustrations. We understand that this review has a focus on sepsis but we suggest that authors check and improve citations and put at least some more graphical material in the manuscript to help the “at a glance” fruition of the paper. One about climate changes and increased V.Vulnifucus diffusion could be one. Another image could deal with V. Vulnificus in sepsis"

Answer: "As suggested by the reviewer, we have increased the number of references cited in our paper and added more figures. However, the number of citations remains lower compared to other reviews on the subject because this paper is specifically focused on sepsis"

 Comment 2 How does sepsis general treatment differ from specific pathogen directed treatment?

Answer: "We have added a sentence on the topic in the discussion section"

Reviewer 2 Report

Comments and Suggestions for Authors

The manuscript submitted for publication to Microorganisms by Candelli et al., titled: "VIBRIO VULNIFICUS – A REVIEW with a special focus on sepsis" is an interesting review that is focusing on vibrato vulnificus bacterium in the context of sepsis.

The review is well written and structured. It is organized logically and is easy to read/follow by the reader. The list of references is extensive and the authors present a significant volume of recent information.

The reviewer would like to bring up only a few points for the further improvement of an already good manuscript.

1. Consider including a PRISMA figure describing the process followed in addition to the text that is included already.

2. Consider adding a brief section of discussion on the relationship between the studied bacterium and the human gut microbiome.

3. Also consider adding a brief section in terms of potential clinical implications/applications, and future directions in research (interesting questions/gaps worthy of investigation as identified by this review). It would be helpful if these sections were clearly delineated in the manuscript.

 Good job overall.

Author Response

Comment 1: "The manuscript submitted for publication to Microorganisms by Candelli et al., titled: "VIBRIO VULNIFICUS – A REVIEW with a special focus on sepsis" is an interesting review that is focusing on vibrato vulnificus bacterium in the context of sepsis. The review is well written and structured. It is organized logically and is easy to read/follow by the reader. The list of references is extensive and the authors present a significant volume of recent information.

Answer: "Thank you for the kind words of appreciation for our paper and for the valuable suggestions that have allowed us to further improve it"

The reviewer would like to bring up only a few points for the further improvement of an already good manuscript.

  1. Comment 1 "Consider including a PRISMA figure describing the process followed in addition to the text that is included already"

Answer: "Although our study is not a systematic review, we have included a PRISMA flow diagram, as kindly suggested by the reviewer"

  1. Comment 2 "Consider adding a brief section of discussion on the relationship between the studied bacterium and the human gut microbiome."

Answer: "a brief section on the relationship between V. vulnificus and gut microbiota has been added as suggested by the reviewer"

  1. Comment 3 "Also consider adding a brief section in terms of potential clinical implications/applications, and future directions in research (interesting questions/gaps worthy of investigation as identified by this review). It would be helpful if these sections were clearly delineated in the manuscript"

Answer: "we add a future research/potential clinical implication has been added as suggested by the reviewer"

Reviewer 3 Report

Comments and Suggestions for Authors

Overall the manuscript appears to be clearly and carefully written. I think that the manuscript might deserve publication in the Journal of microorganisms after some points are dealt with and some missing details are added prior to publication as follows:

1-      The authors should address the novelties and motivations of the current work.

2-      It is necessary to discuss diagnostic methods for V. vulnificus in current manuscript.

3-      Please discuss in the text the Light-based V. vulnificus therapeutic approaches.

4-      In addition to the References list, other current research on the use of laser light is proposed to be reviewed and included if they are beneficial:

Ø    Aquaculture, vol. 561, 738628. https://doi.org/10.1016/j.aquaculture.2022.738628

Ø    Nanomaterials 2022, 12, 3757. https://doi.org/10.3390/nano12213757

Ø    Opt Quant Electron 56, 977 (2024). https://doi.org/10.1007/s11082-024-06781-1

Ø    Lasers Med Sci 39, 144 (2024). https://doi.org/10.1007/s10103-024-04080-5

Ø    Lasers Med Sci 36, 641–647 (2021). https://doi.org/10.1007/s10103-020-03104-0

Author Response

Overall the manuscript appears to be clearly and carefully written. I think that the manuscript might deserve publication in the Journal of microorganisms after some points are dealt with and some missing details are added prior to publication as follows:

Answer: “Thank you for your kind words of appreciation for our review and for the suggestions that prompted us to significantly improve it.”

Comment 1-      The authors should address the novelties and motivations of the current work.

Answer: “We added at the end of the introduction the reason why we think this review is needed: The aim of this review is to provide a novel contribution by focusing specifically on Vibrio vulnificus-induced sepsis, a severe and often fatal condition with an increasing prevalence due to climate change, even in countries where it has not been previously reported. Given its devastating impact on patient outcomes, a focused analysis of its pathophysiology, clinical presentation, and management strategies is essential to inform clinical practice and drive future research.”

Comment 2-      “It is necessary to discuss diagnostic methods for V. vulnificus in current manuscript”

Answer : “As suggested by the reviewer, we have added a paragraph on the diagnostics for Vibrio vulnificus infection.”

Comment 3-      Please discuss in the text the Light-based V. vulnificus therapeutic approaches.

Answer:” We have also expanded, as requested by the reviewer, the section on the treatment of the infection using  light-based terapeutic option”

Comment 4-      “In addition to the References list, other current research on the use of laser light is proposed to be reviewed and included if they are beneficial: 1    Aquaculture, vol. 561, 738628. https://doi.org/10.1016/j.aquaculture.2022.738628 2     Nanomaterials 2022, 12, 3757. https://doi.org/10.3390/nano12213757 3    Opt Quant Electron 56, 977 (2024). https://doi.org/10.1007/s11082-024-06781-1 4    Lasers Med Sci 39, 144 (2024). https://doi.org/10.1007/s10103-024-04080-5  5    Lasers Med Sci 36, 641–647 (2021). https://doi.org/10.1007/s10103-020-03104-0”

Answer: “We have added some of the publications suggested by the reviewer that were most relevant to our review.”

Round 2

Reviewer 3 Report

Comments and Suggestions for Authors

The authors have made reasonable changes to the manuscript in response to my previous suggestions and concerns. In my opinion, the manuscript now contains all the information and is ready for publication as a regular article in the Journal Microorganisms.